# Linked-read sequencing of gametes allows efficient genome-wide analysis of meiotic recombination

Hequan Sun [1,5], Beth A. Rowan[2,5], Pádraic J. Flood[3], Ronny Brandt[4], Janina Fuss[4], Angela M. Hancock [3], Richard W. Michelmore[2], Bruno Huettel [4] & Korbinian Schneeberger [1]

Meiotic crossovers (COs) ensure proper chromosome segregation and redistribute the genetic variation that is transmitted to the next generation. Large populations and the demand for genome-wide, fine-scale resolution challenge existing methods for CO identification. Taking advantage of linked-read sequencing, we develop a highly efficient method for genome-wide identification of COs at kilobase resolution in pooled recombinants. We first test this method using a pool of *Arabidopsis* $F_2$ recombinants, and recapitulate results obtained from the same plants using individual whole-genome sequencing. By applying this method to a pool of pollen DNA from an $F_1$ plant, we establish a highly accurate CO landscape without generating or sequencing a single recombinant plant. The simplicity of this approach enables the simultaneous generation and analysis of multiple CO landscapes, accelerating the pace at which mechanisms for the regulation of recombination can be elucidated through efficient comparisons of genotypic and environmental effects on recombination.

[1] Department of Chromosome Biology, Max Planck Institute for Plant Breeding Research, Carl-von-Linné-Weg 10, 50829 Cologne, Germany. [2] The Genome Center and Department of Plant Sciences, University of California, Davis, 451 East Health Sciences Drive, Davis, CA 95616, USA. [3] Department of Plant Developmental Biology, Max Planck Institute for Plant Breeding Research, Carl-von-Linné-Weg 10, 50829 Cologne, Germany. [4] Max Planck-Genome-Center Cologne, Max Planck Institute for Plant Breeding Research, Carl-von-Linné-Weg 10, 50829 Cologne, Germany. [5]These authors contributed equally: Hequan Sun, Beth A. Rowan. Correspondence and requests for materials should be addressed to B.A.R. (email: browan@ucdavis.edu) or to K.S. (email: schneeberger@mpipz.mpg.de)

During meiosis, existing genetic variation is reshuffled and passed down to offspring through COs that exchange portions of homologous chromosomes. This results in new combinations of alleles in the next generation that can generate novel phenotypic variation, which is the raw material for both natural and artificial selection[1–3]. In most organisms, the locations of COs along chromosomes do not form a random distribution[4–11]. Thus, the local CO rate governs the types of allelic combinations that can arise through sexual reproduction. Both environmental and genetic factors have been shown to affect CO rates and distributions; some examples are the local sequence divergence or rearrangements between the homologous chromosomes[12–14], the chromatin context[15–18], variation in DNA repair mechanisms[19–26], and environmental stress[27–29].

Despite a large body of research on CO formation, our knowledge of what determines where and how often COs occur is still incomplete, in part because the time, effort, and resources needed to study this phenomenon have been limiting. Genotyping recombinant individuals, either by classical methods[30], reduced representation sequencing[31,32], or whole-genome sequencing[33] and performing cytological analysis of meiotic cells[25] represent the common methods for determining CO locations and frequencies. However, none of these methods is easily suited for high-throughput analysis of thousands of COs in parallel. The use of recombination reporters[34,35] and pollen-typing[36] enable rapid screening, but can only assess differences in CO frequency in a specific region of the genome. The availability of an efficient method to assess the genome-wide distribution and frequency of COs at a high resolution would greatly enhance our understanding of the processes that govern meiotic recombination.

Here we investigate the use of linked-read sequencing of bulk recombinants for high-throughput genome-wide determination of COs in *Arabidopsis thaliana*. We first develop and assess this approach using bulked $F_2$ individuals within known recombination sites and then apply this method to hybrid pollen to generate a genome-wide CO map with a single sequencing experiment and without even growing a single recombinant plant. These results show that the time and effort needed to generate genome-wide CO maps can be reduced to sequencing and analyzing a single DNA library, making it feasible to compare multiple CO maps and thereby determine the effect of genetic and environmental factors within a single study. We believe that this method will be widely applicable to different organisms and will have a sizeable impact on the design of experiments aiming to decipher how meiotic recombination is regulated.

## Results

**CO breakpoint detection from bulk recombinants**. To establish a set of COs for verifying our method, we first performed whole-genome sequencing of 50 individual $F_2$ plants derived from a cross of two of the best-studied inbred lab strains of Arabidopsis, Col-0 and L*er* and used the haplotype reconstruction software TIGER[33] to determine a benchmark set of 400 COs across all 50 genomes (Fig. 1; Methods; Supplementary Data 1 and 2).

We then bulked the identical 50 $F_2$ plants by pooling individual leaves of comparable size and extracting high molecular weight (HMW) DNA[37] (Fig. 1). After size selection and quality control (Supplementary Fig. 1), we loaded 0.25 ng DNA into a 10X Genomics Chromium Controller. The Chromium Controller encapsulates millions of gel beads as GEMs (Gel bead in EMulsion), each of which is loaded with a small number of long DNA molecules. These long molecules are fragmented and ligated with GEM-specific DNA barcodes to generate a 10X library suitable for Illumina sequencing. This library, which we called

P50L25, was whole-genome sequenced with 84 million 151 bp-read pairs (Supplementary Data 1).

After aligning the reads against the Col-0 reference sequence[38] using *longranger* (v2.2.2, 10X Genomics), we recovered 3.6 million molecules (≥1 kb) including 116 million reads using a newly developed computational tool, *DrLink*, which can be downloaded at https://github.com/schneebergerlab/DrLink (Fig. 2a; Methods). On average, these molecules identified by *DrLink* were ~45 kb in size and were covered by ~21 read pairs, leading to a molecule base coverage of ~0.16× (Fig. 2b–d; Supplementary Data 1). To avoid chimeras resulting from the accidental co-occurrence of two independent, but closely spaced molecules with identical barcodes, we selected molecules which were smaller than 65 kb that had fewer than 55 reads and no heterozygous genotypes. These thresholds were based on the distributions in molecule size and read number per molecule (Fig. 2b–d; Methods; Supplementary Note 1). Overall, 2.7 million molecules passed all filtering.

Initially, we genotyped these molecules using *DrLink* at the ~660,000 SNP markers predicted by 10X Genomics' *longranger* software. If an individual molecule was composed of two distinct clusters of different parental alleles and fulfilled additional criteria regarding length and marker distribution (Fig. 2a; Methods; Supplementary Data 3; Supplementary Note 1), the molecule was considered as a recombinant molecule revealing a CO breakpoint. Using the SNPs called by *longranger*, we predicted 1786 recombinant molecules with a median CO breakpoint resolution (distance between the two flanking markers) of 6.7 kb. However, a comparison with the 400 benchmark COs revealed that only 674 of the molecules overlapped with verified COs, while the remaining 1112 were putative false positives (FP). Many of the FPs appeared close or within structural rearrangements between the parental genomes, suggesting that the molecule reconstruction using the markers provided by *longranger* is vulnerable to unrecognized structural differences between the parental genomes and thereby leads to false predictions of recombinant molecules (Supplementary Note 1).

Making use of chromosome-level assemblies of both parental genomes[38,39], we defined a new marker set composed of ~500,000 SNP markers, which only included SNPs in non-rearranged, co-linear regions between the parental genomes (Supplementary Data 4). We repeated the analysis with this new marker set and filtered for molecules that appeared recombinant independent of which parental genome was used as reference sequence. The *DrLink* analysis identified only 558 recombinant molecules, and 475 (85.1%) of the CO breakpoints among these molecules (median resolution 5.3 kb) overlapped with the benchmark COs. The remaining 83 (14.9%) COs were putative FPs most likely due to the co-occurrence of different molecules with the same barcode (Supplementary Note 1). However, some of them might have even been true recombination events that were missed in the benchmark set (e.g., two closely spaced COs, gene conversion events[33,40] or mitotic COs, which were only present in some cells of the sequenced material).

**Increasing the number of molecules per library**. To test whether the number of (recombinant) molecules per library would be increased by increasing the DNA loading, we generated two additional 10X libraries from the same DNA pool by loading 0.40 ng (P50L40) and 0.75 ng (P50L75) into the Chromium Controller (Fig. 1; Methods). We sequenced these libraries with 104.8 and 212.3 million read pairs in anticipation of an increased number of molecules within both libraries (Fig. 2b–d). Following the same analysis as for the first library, the higher DNA loading drastically increased the number of recovered molecules. As compared to the

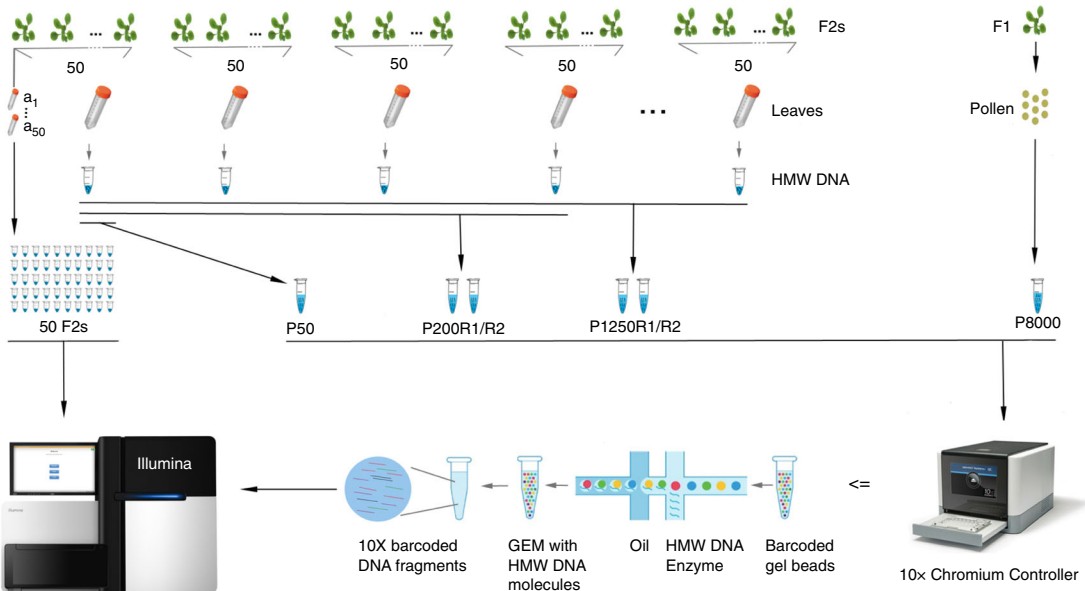

**Fig. 1** Experimental design and CO detection using linked-reads. $F_1$ and $F_2$ plants were derived from crosses of two divergent *A. thaliana* accessions. *Leaf sampling*: Leaves of 50 selected $F_2$s were individually sampled ($a_{1-50}$). In addition, the leaves of all other plants were pooled in batches of 50 $F_2$s (for further merging), where the 50 individually sampled plants formed one of the pools. Also, pollen from a single $F_1$ plant was sampled. *DNA extraction*: DNA from individual the samples $a_{1-50}$ were extracted for Illumina whole-genome sequencing, while the DNA of the 50-$F_2$ and pollen samples were extracted with a protocol for high molecular weight DNA (see Methods and Mayjonade et al. [37]). Pooling: Four and twenty-five 50-$F_2$ samples were merged leading to 200-$F_2$ and 1,250-$F_2$ pools (P200 and P1250), each with two replicates (R1 and R2). The 50-$F_2$ pool composed of the 50 $F_2$s that were individually sequenced was labelled as P50, while the pollen pool was labelled as P8000. *Library preparation*: 50 individual $F_2$ DNA samples were used for preparing Illumina DNA TruSeq libraries. The P50 DNA sample was loaded into a 10X Chromium Controller (illustration modified from 10X Genomics with official permissions) with three different amounts of DNA including 0.25 ng, 0.40 ng and 0.75 ng (P50L25, P50L40, and P50L75). P200R1/R2 and P1250R1/R2 were loaded using 0.75 ng and P8000 was loaded using 1.00 ng. Sequencing: All libraries were sequenced on Illumina HiSeq3000/4000 sequencers

2.7 million molecules for P50L25, we now found 4.7 and 10.5 million molecules for P50L40 and P50L75 after filtering (Table 1). This also increased the number of recombinant molecules to 1012 in P50L40 and 2519 in P50L75 as compared to 558 in P50L25. This, however, had the cost of also increasing the FP rate by 5% in P50L40 and 10% in P50L75 (Table 1).

To investigate the effect of the FPs, we compared the distribution (i.e., recombination landscape) of all 2519 COs in the library with the highest FP rate, P50L75, with two other landscapes: one calculated only from 1874 true COs (those overlapping with the 400 benchmark COs) and one calculated only from the remaining 645 false recombinant molecules (Methods). The recombination landscape calculated for all COs (TP + FP, i.e., the set of TP and FP COs) was nearly identical to the landscape generated from true COs only (TP), while the distribution of the FPs was highly random (Fig. 3a; Supplementary Fig. 2: K–S test, *p*-value = 9.6e-01), suggesting that FPs hardly obscure true recombination landscapes. In fact, correlating the sliding window-values of each of the three CO landscapes revealed that the landscape calculated from all CO sites was almost perfectly correlated to the one generated from real COs (Fig. 3b: Correlation test, Pearson's *r* 0.97, *p*-value < 2.2e-16), while the frequency of FP along the chromosomes was not correlated to the frequency of real COs and was only marginally correlated to landscape of all COs.

**Association of COs with genomic features**. To test the association of COs with genomic features, we checked all of the CO sites for their annotation in the Col-0 reference sequence (Fig. 3c). In comparison to randomly placed CO sites, permutation tests showed that the 1874 true COs sites were significantly enriched in promoter regions (*p*-value 5.0e-03) and intergenic

regions (*p*-value 1.4e-03) and were significantly depleted in gene bodies (*p*-value 6.0e-04) and transposable elements (*p*-value 9.0e-04) recapitulating regional preferences, which have been described before[40,41]. When additionally including the FP, the CO set showed the same significant regional associations with one exception: COs were slightly, but significantly enriched compared to random COs at gene ends. Gene ends represented only a minor fraction of all COs in both datasets and the difference between the two was marginal (5.16% within TP + FP CO sites vs. 4.90% within TP COs sites) and these percentages were much closer to each other than the percentage in the randomized CO dataset (4.40%). A later analysis with a larger CO dataset helped to avoid this spurious enrichment (see next section). To examine CO associations with genomic features at a finer scale, we analyzed the regional preferences of the CO sites in individual transposable element super families because these were found to have strong differences in CO rates[42]. Though this only included 331 putative CO sites, we found that COs were significantly depleted within *LTR-Gypsy* and *En-Spm* super families and enriched within *Helitron* and *LINE* elements, as has been previously shown[42], whether or not FP were included (Supplementary Fig. 3). In addition, the presence of FP did not affect the previously reported relationships between COs and GC-content or DNA methylation[40]. The two datasets showed nearly identical correlations between CO frequency and GC-content (Fig. 3d: Correlation test, Pearson's *r* −0.46 and −0.45, both *p*-value < 2.2e-16). Similarly, COs in both datasets were found in regions with low levels of methylation (Fig. 3e).

Together this suggests that our method is not only effective when facing an increasing amount of FP in libraries with a large number of molecules, but that it is also powerful enough to accurately identify chromosome-wide and local CO patterns.

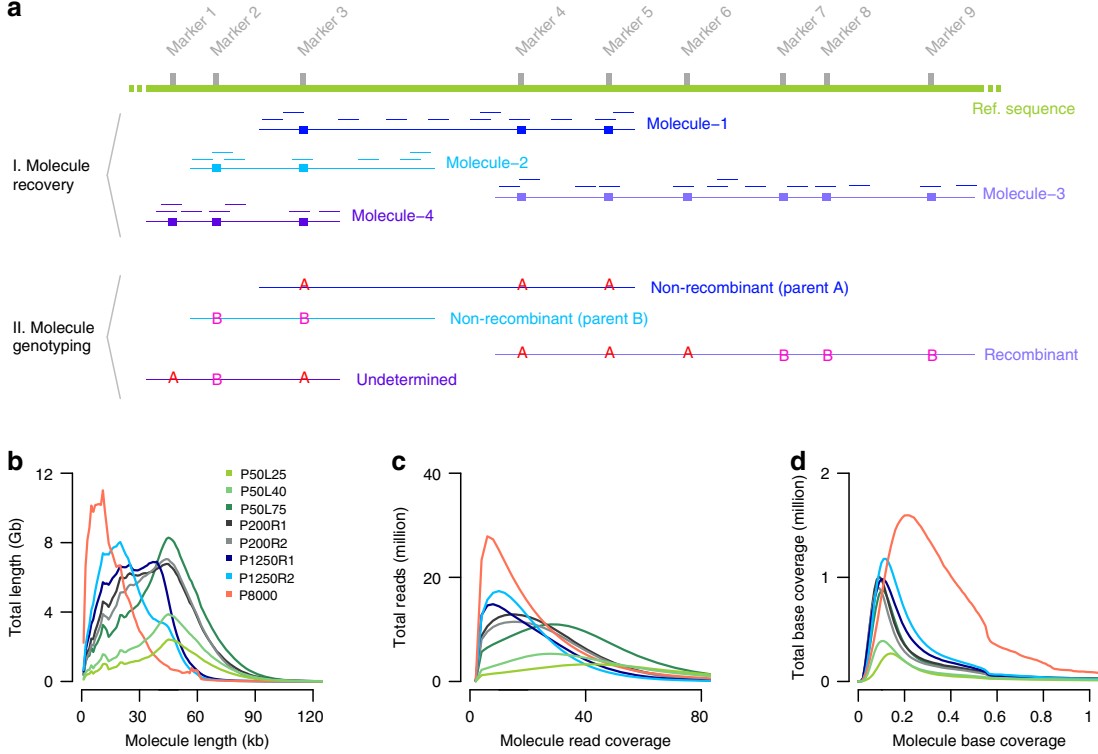

**Fig. 2** Molecule recovery and genotyping using *DrLink*, and molecule characteristics. **a** Reads with the same barcode (represented by short lines in matching colors) that were aligned within close proximity (less than 25 kb apart) have been connected to recover molecules (Molecule-1 to Molecule-4). The genotypes that can be assessed with the read alignments of each recovered molecule fall into three major categories, non-recombinant (Molecule-1, Molecule-2), recombinant (Molecule-3), and undetermined (Molecule-4). **b** Length of recovered molecules for each of the 10X libraries. **c** Number of reads per recovered molecule. **d** Recovered molecule base coverage. Source Data are provided as a Source Data file Source_Data_main_Figure_2.zip

| Table 1 Effect of increasing the molecule number by increasing DNA loading | | | |
|---|---|---|---|
| **Pool** | **P50L25** | **P50L40** | **P50L75** |
| Raw molecules (≥1 kb)[C] | 3,577,104 | 5,804,559 | 12,551,833 |
| Raw molecules (≥1 kb)[L] | 3,462,796 | 5,631,294 | 12,240,762 |
| Filtered molecules[C] | 2,664,977 | 4,783,828 | 10,651,147 |
| Filtered molecules[L] | 2,565,403 | 4,642,515 | 10,417,489 |
| Total CO molecules[CL] | 558 | 1012 | 2519 |
| TP CO molecules[CL] | 475 | 804 | 1874 |
|  | (85.1%)[R] | (79.4%)[R] | (74.4%)[R] |
| TP unique (non-redundant) COs[CL] | 254 | 325 | 363 |
|  | (63.5%)[U] | (81.3%)[U] | (90.8%)[U] |
| FP molecules[CL] | 83 | 208 | 645 |
|  | (14.9%)[R] | (20.6%)[R] | (25.6%)[R] |

*TP* true positive, *FP* false positive
[C]Number based on Col-0 reference genome[38]
[L]Number based on Ler reference genome[39]
[CL]Number based on intersection of two sets of CO predictions (of C and L)
[R]Percent CO molecules of all molecules
[U]Recall rate

**Increasing the number of genomes per library**. Many of the recombinant molecules that we found in the pool had identical CO breakpoints. This implied that we re-discovered some of the CO breakpoints in independent molecules, suggesting that there were more recombinant molecules in the library than independent CO breakpoints in the pooled genomes. For instance, there were only 363 distinct COs recovered by the 1874 recombinant molecules in P50L75 (Table 1). Even though identifying a single CO breakpoint multiple times can help to increase its resolution and reliability, it does reduce the overall number of distinct COs that can be found with one library.

The number of recombinant molecules that can be identified within one 10X library greatly depends on the number of molecules in the library. As the probability to overlap with a CO breakpoint is theoretically the same for each molecule, the number of CO molecules increases linearly with the number of analyzed molecules. The number of molecules heavily relies on the amount of DNA that is loaded in the 10X Chromium machine (while sequencing depth has only a marginal effect on molecule recovery). As a consequence, each 10X library includes an almost fixed number of recombinant molecules.

In turn, the inclusion of more plants in the pool does not increase the number of recombinant molecules. However, the inclusion of more plants does increase the number of independent recombinant molecules and thereby increases the number of distinct CO events found with one library (Supplementary Note 2). While this maximizes the number of distinct CO breakpoints that can be found with one library, it also implies that a majority of the breakpoints that are in the pool will not be found. Thus, pools with a small number of plants/CO breakpoints (smaller than the number of recombinant molecules) will reveal all CO breakpoints with high confidence; pools with a large number of plants/CO breakpoints will reveal a large number of distinct CO breakpoints, but cannot achieve the identification of all CO breakpoints. Hence, the false negative rate is mostly determined by the number of independent genomes in a pool and much less by the actual detection success of recombinant molecules.

To test the effect of genome number on the number of distinct COs in practice, we applied our method to pools of 200 and 1250 $F_2$ plants. We generated two individual 10X libraries each for a pool of 200 (P200R1, P200R2) and a pool of 1250 (P1250R1 and P1250R2) different $F_2$ plants (Fig. 1; Supplementary Figs. 1 and 4;

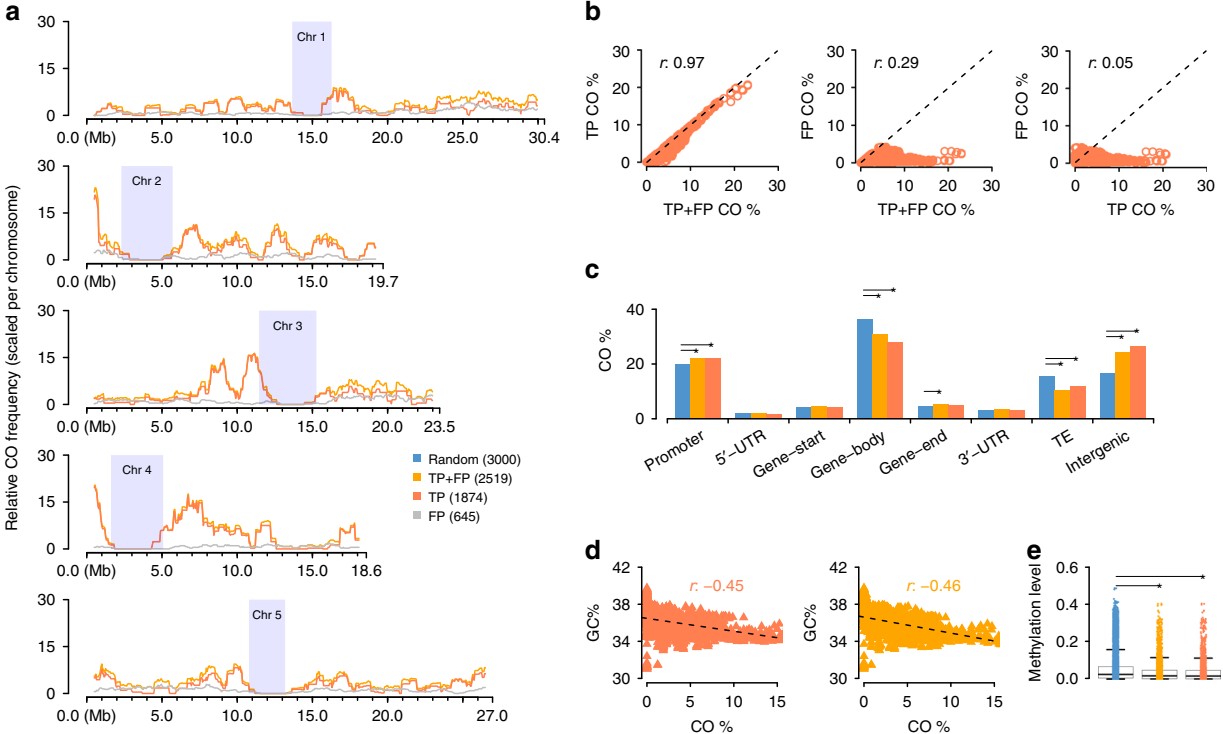

**Fig. 3** Genome-wide CO landscape formation and feature association. **a** Sliding window-based (window size 1 Mb, step size 50 kb) recombination landscapes calculated for true positives (TP), false positives (FP) and the combined set of TP + FP recombinant molecules of P50L75. Heterochromatic regions[44] are indicated by rectangles in light blue. **b** Correlation tests of regional CO frequencies comparing TP vs. TP + FP (Pearson's r 0.97), FP vs. TP + FP (Pearson's r 0.25) and FP vs. TP (Pearson's r 0.05). **c** Association of the TP and TP + FP sets with different genomic features in contrast to the random expectation (permutation test). **d**. Correlation tests of regional GC-content with CO frequency within the TP and TP + FP sets. **e** Association of TP or TP + FP sets with DNA methylation, compared with a random expectation (t-test; Methods). Note: in **c** and **e**, the expectations were obtained based on randomly sampled intervals within the reference genome excluding heterochromatic regions. The middle position of a CO interval was used for associations with genomic features in **c**, where the promoter of a gene was defined as the 1000 bp region upstream of the transcription start site, and the gene start/end as the first/last 200 bp of a gene. For each feature, a permutation test (oneway test in R coin package) was performed between the TP set (or TP + FP set) with 1,000 sets of randomly sampled 3000 intervals. The asterisks indicate the observed values (either from the TP or the TP + FP set) were significantly different from a random expectation. Source Data are provided as a Source Data file Source_Data_main_Figure_3.zip

Methods). These libraries were sequenced with 168.2–194.4 million read pairs (Supplementary Data 1) and revealed 12.3–16.0 million molecules. For P200R1 and P200R2, we found 2538 and 2334 recombinant molecules, which identified 1779 (69.5%) and 1606 (68.8%) distinct CO sites. For the pool of 1250 plants, there were 2880 and 3430 recombinant molecules, which uncovered 2386 (82.8%) and 2788 (81.3%) distinct CO sites. Comparison of these COs with 3320 COs determined using individual whole-genome sequencing for an independent set of 437 Col x Ler F$_2$s[20,21] showed consistent genome-wide patterns, including on identifying genomic regions with relatively high CO occurrences (Supplementary Fig. 5). Although we were unable to assess FP rate in these larger pools, the above consistencies in feature associations (even at gene ends with ~5% COs) suggested that FPs do not interfere with the proper identification of true CO patterns. While the larger pools revealed more distinct CO sites, there were still nearly 20% COs overlapping, but it does not necessarily imply that these are not unique CO events. As the genome size of *A. thaliana* is relatively small (~135 Mb)[43], independent COs have an unneglectable probability to overlap (Supplementary Note 2; Supplementary Data 5). According to simulations, given a pool of 1250 F$_2$s, ~15% of the independent CO events are expected to overlap (Supplementary Data 5 and 6) making up for large parts of the overlapping recombinant molecules in the 1250 pools.

**Estimating relative recombination frequency**. Since the probability to identify a CO breakpoint within a molecule depends on the average recombination frequency among the pooled genomes, it might be possible to estimate the average recombination frequency from the fraction of observed recombinant molecules. However, as the probability to identify a CO breakpoint also depends on the length and sequencing coverage of the molecules within a library, it is only meaningful to calculate an average recombination frequency that is relative to the actual molecule characteristics (or relative recombination frequency). To test for significant differences in relative recombination frequencies between libraries, it is therefore essential that they have close-to-identical molecule characteristics. For libraries with differences in molecule length and sequencing coverage distributions, subsampling can be used to generate such identical distributions. Once the distributions are similar, the relative recombination frequency can be determined as the number of recombinant molecules per million molecules ($C^M$). Repeated subsampling even allows for the calculation of confidence intervals for $C^M$ values and thereby allows for testing significant differences between the average recombination frequencies of different pools.

First, we compared the CO frequencies of P200R1 and P200R2, which were two independent libraries generated from the same DNA (Fig. 1; Fig. 2b–d; Methods). After subsampling (Supplementary Fig. 6), there was no significant difference in $C^M$ values

**Table 2 Relative recombination frequency in F₂ and pollen pools**

|  | Pool | Molecules[a] | CO molecules[b] | $C^M$ |
|---|---|---|---|---|
| F₂ | P200R1 | 4,845,588 ± 1,556 | 129 ± 7 | 26.7 ± 0.4 |
| F₂ | P200R2 | 4,856,886 ± 1,371 | 130 ± 7 | 26.8 ± 0.4 |
| F₂ | P1250R1 | 4,650,420 ± 1,662 | 126 ± 9 | 27.2 ± 0.6 |
| F₂ | P1250R2 | 4,309,936 ± 1,454 | 117 ± 10 | 27.1 ± 0.7 |
| Pollen | P8000 | 4,902,994 ± 1,722 | 212 ± 10 | 43.2 ± 0.6 |

[a, b]The values are $\mu \pm \sigma$, where $\mu$ is the mean and $\sigma$ is the standard deviation of the molecule and CO numbers in 50 random sub-samplings. For $C^M$, i.e., the ratio of CO molecules to total molecules scaled by a factor of $10^6$, the values are $\mu \pm s$ giving 95% confidence intervals for the mean $\mu$, where $s$ is $2.0096 \times \sigma / 50^{0.5}$ with $\sigma$ being the standard deviation. Source Data are provided as a Source Data file Source_Data_main_Table_2.zip

(26.7 ± 0.4 and 26.8 ± 0.4), as expected for these libraries (Table 2). We repeated this comparison for P1250R1 and P1250R2, which differed greatly in their original molecule characteristics (Fig. 2b–d). After subsampling to the same molecule distributions as for the smaller pools (Supplementary Fig. 6), the $C^M$ values of 27.2 ± 0.6 and 27.1 ± 0.7 were also not significantly different from each other (Table 2). Moreover, when comparing the 200 and 1250 recombinant pools, we also found no significant difference between any of the pools (all confidence intervals overlapped), which is expected, given that all libraries were generated from individuals of the same F₂ population. Together, these results show that $C^M$ values are stable against differences in the molecule characteristics and pool sizes and allow for determining and comparing average recombination frequencies between samples.

**Estimating CO frequency and landscapes from gametes.** Crosses between divergent strains provide the simplest opportunity for determining COs, but typically require the generation of a recombinant population after a single round of meiosis in a hybrid context. Observing COs directly in the products of meiosis (gametes) would greatly expedite the study of recombination, especially in inbred species with long generation times. To test how our method performs on recombinant gametes, we extracted HMW DNA from Col x Ler F₁ hybrid pollen, created a 10X library (P8000) and sequenced it with 319.9 million read pairs (Fig.1; Supplementary Fig. 1f; Supplementary Data 1). Following the same analysis as for the F₂ pools, we identified 20 million molecules with an average molecule base coverage of 0.26× (Fig. 2b–d). Among those, there were 3246 recombinant molecules with a median CO breakpoint resolution of 8.0 kb.

We compared relative recombination frequencies in pollen with the P1250R1 and P1250R2 pools after re-sampling molecules to obtain comparable characteristics (Supplementary Fig. 6; Methods). The CO frequency estimate ($C^M$) for pollen was significantly higher than for F₂s (Table 2), consistent with the higher male recombination rate in this species[44].

The distributions of the CO breakpoints in pollen and F₂s were highly correlated in most regions of the genome (Correlation test, Pearson's $r$ 0.80~0.86, both $p$-value < 2.2e-16), and exhibited only 13 regions with local differences across the entire genome (Fig. 4; Methods). Though we cannot exclude random processes, these differences provide the starting point to investigate whether there are post-meiotic processes that influence which gametes contribute to the next generation.

We further compared the recombination landscape of our pollen sample to a previously published recombination landscape from a Col-0 x Ler backcross population, where all COs were derived from male meiosis[44]. This dataset, consisting of 7418 COs at an average resolution of ~320 kb, was generated from 1505 individuals genotyped with 380 SNP markers that were evenly spaced throughout the genomes. After binning our CO data into the same windows, we found that the recombination landscapes were broadly similar (Supplementary Fig. 7: Correlation test, Pearson's $r$ 0.41,

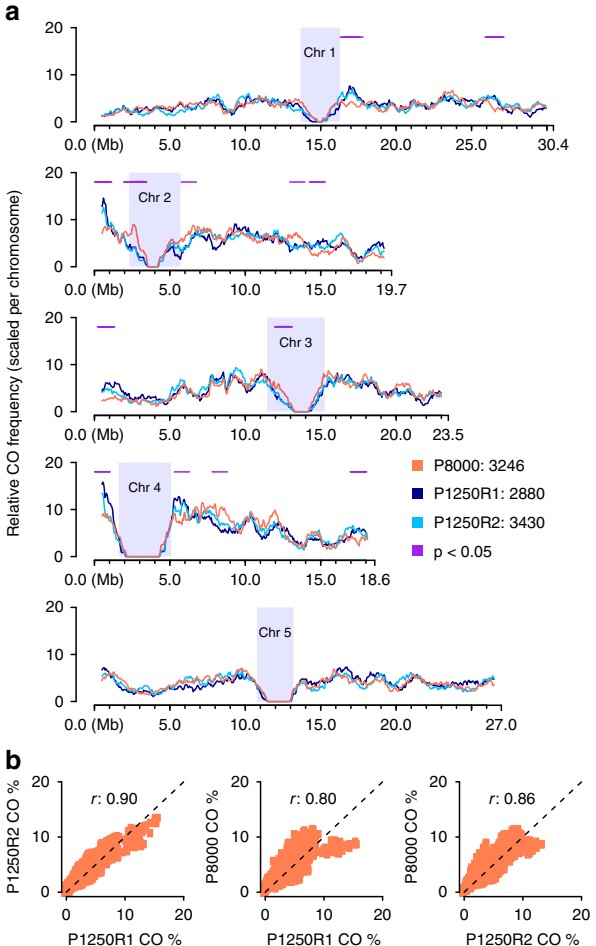

**Fig. 4** Genome-wide recombination landscape in pollen. **a** Comparison of sliding window-based (window size 1 Mb, step size 50 kb) recombination landscapes in pollen and F₂ populations (P1250R1/R2), showing genome-wide consistencies but with 13 regional differences (1.1 ± 0.2 Mb) as indicated by horizontal lines in purple with $p < 0.05$ (i.e., $p$-values of *Fisher's* exact test; Methods). Heterochromatic regions[44] are indicated by rectangles in light blue. **b** Correlation of genome-wide CO frequencies in pollen and F₂s (Correlation test, Pearson's $r$ 0.80-0.86, both $p$-value < 2.2e-16). Source Data are provided as a Source Data file Source_Data_main_Figure_4.zip

$p$-value < 2.2e-16). Unexpected differences, however, were observed mostly at the end of chromosomes. COs formed by male meiosis and identified in recombinant BCF₁ plants showed significant increases at the end of some chromosomes[44], while most of these increases were not observed in pollen data.

In general, this shows that a single sequencing library is sufficient to derive genome-wide CO patterns from pooled gametes without the need for producing any recombinant individuals. Such CO maps have the advantage to measure sex-specific meiotic recombination without doing any backcrosses and are less influenced by post-meiotic biases.

## Discussion

For the past century, the study of recombination has relied on inferring or determining the genotypes at (limited) marker positions along the chromosomes of recombinant individuals. Here, we developed an efficient and accurate method for detecting recombination breakpoints in pooled DNA using linked-read sequencing. This method can be applied directly to gametes, avoiding the need for generating or genotyping recombinant populations. In turn, this simplification allows for the production of multiple different genome-wide CO maps with relatively little effort. Although it has long been known that many environmental factors influence CO rates, such as temperature, water stress, salinity, and pathogen stress[45,46], the genetic mechanisms underlying this sensitivity have not been well characterized. We expect that our approach will greatly aid research in this area, as it is expected to make it feasible to directly compare CO maps generated from many different genotypes or environments. Long-read sequencing or single-cell genome sequencing of individual pollen would be two alternative methods, but the current costs of library preparation and sequencing or time and effort needed for sample processing are prohibitive for comparing multiple samples. For example, Nanopore and PacBio sequencing require 8–70 times the cost of 10X linked-read sequencing to obtain a similar number of recombinant molecules (Supplementary Data 7). Published methods for applying single-cell genome sequencing to the identification of COs incur the high labor cost of sorting of individual cells or nuclei[47,48]. The resolution of COs is also much poorer for single-cell genome sequencing for the same sequencing cost as 10X linked-read sequencing, given that much higher coverage is needed to achieve kb-scale resolution.

The small genome size of the species we used in this study, *A. thaliana*, was potentially an obstacle to accurately scoring CO events using linked-read sequencing, as this technology assigns the same barcode to a small number of individual DNA molecules that are tens of kb in length. Since this process is random, two molecules that originate from different genomes in the pool that happen to align near each other can receive the same barcode. The probability of these collision events is much higher in small genomes and can mimic COs if the two molecules have different parental genotypes, giving rise to false positives (FP). However, the FP had little effect on the overall accuracy of the CO predictions (Fig. 3). Even though the FP rate did not pose a substantial problem, making several libraries using a reduced input would result in high molecule numbers at the lower FP rate. This would incur higher costs of library production while maintaining the same sequencing cost. Moreover, when applying this method to larger genomes, the chance of collision of two molecules with the same barcode would be lower, leading to a lower likelihood of FPs.

The spatial distributions of COs identified with our method recapitulated known recombination landscapes and precisely co-localized with genomic and epigenetic features that have been reported to be associated with meiotic recombination[36,39,41]. The resolution of the CO breakpoints was very precise, with the median CO interval less than 10 kb in each of the samples. This accuracy, however, could only be achieved by filtering the marker list for co-linear regions between the parental genomes as genomic rearrangements between the parents were hotspots for FPs.

The careful validation studies of our method using $F_2$s provided the basis for determining the COs directly in gametes. Linked-read sequencing of bulk $F_1$ pollen led to the discovery of ~3500 CO events that showed a similar distribution as was recently published for male meiosis (Supplementary Fig. 7). Dreissig et al. previously assessed the recombination rate in barley pollen using whole-genome sequencing of individual pollen nuclei[47]. While this allowed for the analysis of CO interference, their method employed a more technically challenging library preparation that does not scale to high numbers and achieved a much lower CO resolution.

Interrogating COs in gametes (pollen, in the case of plants or sperm in animals) has many advantages over current approaches. It avoids the laborious and time-consuming step of growing/rearing recombinant populations. It reduces the number of generations required, greatly facilitating recombination studies in inbred organisms with long generation times. Because entire recombination landscapes can be generated from single libraries, a recombination landscape can now be studied as a single trait. Multiple CO landscapes that can be replicated either in the same or different genetic backgrounds and environments and compared in a single study. In consequence, this now allows for a more complex and sophisticated experimental design to test hypotheses regarding the regulation of recombination.

## Methods

**$F_2$ DNA extraction and library preparation.** $F_2$ seeds from Col-0 x L*er*-0 were stratified for 7 days at 4 °C, sown on soil in 24-pot trays, and grown under 16 h light, 8 h dark cycles at 20 °C for three weeks. DNA pools of up to 1250 $F_2$ individuals were constructed (Fig. 1). HMW DNA was extracted from pools with 50 distinct $F_2$ plants[37]. One 50-$F_2$ pool was selected for validating the method, for which DNA was also extracted from each individual ($a_{1-50}$) and WGS libraries were prepared using the Illumina DNA TruSeq protocol. The DNA of four 50-$F_2$ pools with similar fragment size distribution (Supplementary Fig. 4a) were merged based on equal concentration, from which replicates P200R1 and P200R2 were obtained. In addition, 25 50-$F_2$ pools with divergent fragment size distributions (Supplementary Fig. 4b) were merged based on equal molarity of molecules between 42–70 kb according to FEMTOpulse, AATI genomic DNA quality check, and the resultant DNA was used for replicates P1250R1 and P1250R2.

Size selection was performed on each DNA pool using the Sage Science BluePippin high-pass protocol (i.e., 0.75% Agarose Dye-Free/0.75% DF Marker U1 high-pass 30–40 kb vs3) with starting point at 40 kb to ensure that most molecules were over 40 kb (Supplementary Fig. 1a-e). For P200R1/2 and P1250R1/2, the size-selected DNA was evaluated using the Qubit fluorometer and TapeStation analyzer. After selection, for each library 0.75 ng DNA were loaded into the 10X Chromium controller (Supplementary Data 1). P50 was subjected to the same quality control measures. For this library three different amounts of DNA (0.25, 0.40, and 0.75 ng) were loaded into the 10X Chromium Controller, generating the P50L25, P50L40, P50L75 libraries.

**Pollen DNA extraction and library preparation.** Col *CEN3 qrt 420* and L*er* seeds were stratified for 4–7 days at 4 °C before sowing on soil in 18-pot trays and growing under 16 h light, 8 h dark cycles at 20 °C until flowering. Pollen from a single L*er* plant was used to pollinate a single Col *CEN3 qrt 420* stigma to generate $F_1$ plants. To obtain pollen and extract DNA, we adapted a method from Drouaud and Mezard[49] as follows: Inflorescences were collected from a single $F_1$ plant and ground in 1 mL of 10% sucrose using a mortar and pestle. 9 mL of 10% sucrose were added to the mortar slurry and the resulting homogenate was mixed by pipetting up and down with a wide bore 1 mL pipet tip and filtered through an 80-µM nylon mesh before centrifugation at 350 × g for 10 min at 4 °C. The supernatant was discarded and the pollen pellet was washed two times with 10% sucrose. The pellet was resuspended in four volumes of lysis buffer (100 mM NaCl, 50 mM Tris–HCl (pH 8)), 1 mM EDTA, 1% SDS and proteinase K was added to achieve a concentration of 20 µg/mL. Five to ten 2-mm glass beads were added to the sample and it was vortexed at full speed for 30 s. To check for pollen disruption, a 1-µL sample was removed and mixed with 10 µL of lysis buffer and examined under a microscope. During this time, we verified that the pollen sample was free of large amounts of cell debris. The sample was then vortexed for 30 s and checked for pollen disruption. An equal volume of Tris-saturated phenol was added and the sample was placed on a rotating wheel at room temperature for 30 min. After centrifuging at 15,000 × g for 10 min, the supernatant was transferred to a new tube and mixed with an equal volume of 24:1 chloroform:isoamylalcohol and homogenized by shaking the tube. The tube was centrifuged again at 15,000 × g for 10 min and the supernatant was mixed with 0.7 volumes isopropanol in a new tube

and inverted gently. After another centrifugation step at $15,000 \times g$ for 10 min, the supernatant was discarded and the pellet was washed with 1 mL of 70% EtOH. After a final centrifugation at $15,000 \times g$ for 2 min, the supernatant was discarded and the DNA pellet was allowed to dry at room temperature. The pellet was resuspended in 50 μL of 10 mM Tris-HCl, pH 8 with 0.1 mM EDTA and stored at 4 °C.

DNA was analyzed by field inversion gel electrophoresis to confirm that most molecules were around 48 kb (Supplementary Fig. 1f) before preparing the linked-read library (load: 1.00 ng) using the Chromium Genome Reagent Kit, the Chromium Genome Library Kit & Gel Bead Kit, Chromium Genome Chip Kit v2, and Chromium i7 Multiplex Kit according to the manufacturer's instructions. As 1.00 ng is equivalent to $10^6$ Mb, the haploid genome size is ~135 Mb[43], and each pollen grain is composed of three cells, there were ~3000 pollen grains expected with unique recombinant genomes.

In total, eight 10X and 50 standard Illumina libraries were sequenced on HiSeq 3000/4000 with 151 bp paired-end reads, where the individual $F_2$s were at ~5x and the pools of $F_2$s and pollen nuclei were at $173-659 \times$ (Supplementary Data 1).

**Molecule recovery and genotyping using *DrLink*.** Col-0 and L*er* reference genomes were indexed using the function *mkref* of *longranger* (v2.2.2 10X Genomics). Linked-reads of each sample were aligned against the Col-0 and L*er* reference genomes separately using *longranger wgs* with options:–id = ALIGN-ID–reference = MKREF-INDX-FOLDER–fastqs = READ-PATH–sample = READ-ID–localcores = 40–localmem = 192–noloupe–sex = male–vcmode = freebayes–library = LIBRARY-ID and all other options under default settings (note that option–sex was set as male, which is required by the tool but not affecting the alignment here). Read alignments with the same read barcodes were clustered to recover molecules by *DrLink molecule* function according to their genomic location ensuring that neighboring read alignments were not more than 25 kb away from each other. The resultant molecules (over 1 kb) were further filtered for falsely merged molecules by removing very long or densely covered molecules (Supplementary Note 1). Next, with a set of SNP markers and the barcoded read alignments (in VCF) given by *longranger*, molecules were genotyped by *DrLink recombis* function to identify recombinant molecules, with options for filtering less confident predictions (Supplementary Data 3).

**Generating the CO benchmark set.** For each individually sequenced $F_2$, read alignments against the Arabidopsis Col-0 reference sequence[38] and variant calling were performed using *Bowtie2*[50] (version 2.2.8) and *SAMtools/BCFtools*[51] (version 1.3.1). *TIGER*[33] was used to reconstruct the parental haplotypes using SNPs in co-linear regions between the Col-0 and L*er*[39] genomes. We determined an average of 8.3 COs per diploid genome, totaling 415 COs with a median breakpoint resolution of 0.5 kb (Supplementary Data 2). Of those, 15 COs were located in regions with an inter-marker distance of more than 10 kb. As local CO breakpoint identification relies on markers near the breakpoint, we excluded these COs, generating a final benchmark set of 400 COs.

**Estimating chromosomal CO landscapes.** Recombination landscapes were estimated using sliding windows along each chromosome (window size 1 Mb, step size 50 kb). Within a given window, the CO frequency $f$ was calculated by $f = n / t_i$, where $n$ is the number of COs in the window and $t_i$ is the total CO number within the respective chromosome $i$ across each of the focal libraries. This ensures that different CO landscapes can be compared to each other within the given chromosome.

**Comparison of CO distribution in pollen and $F_2$ populations.** Local CO occurrence in pollen (P8000) and $F_2$ populations (two libraries: P1250R1/2) were compared based on sliding windows (window size 1 Mb, step size 50 kb). Specifically, for each window, we counted both recombinant molecules $R^x$ and non-recombinant molecules $N^x$, where $x$ was either P1250R1/2 or P8000. Then for each window, two *Fisher*'s exact tests were performed between ($R^{P8000}$, $N^{P8000}$) and ($R^{P1250R1/2}$, $N^{P1250R1/2}$). Finally, the windows with both $p$-values $< 0.05$ were identified as different between P8000 and P1250R1/2, and neighboring windows were connected into larger regions.

**Relative CO frequency estimation.** For frequency estimation, we first subsampled molecules from the pools of interest by intersecting their molecule characteristic distributions. Using the molecule length distributions as the basis (Supplementary Fig. 6a), the molecules were randomly sampled down within 1 kb large bins. For each bin, the number of randomly selected molecules per pool was equal to the lowest number of molecules across all pools within this specific bin. To increase the subsampling rate, 80% of the sampled molecules were further randomly selected within each bin. After this second subsampling, the read and base coverage distributions were also highly similar (Supplementary Fig. 6b, c). Final molecule filtering was applied following the same strategy as used for the complete sets of molecules, by applying 30 kb as size and 24 as read number thresholds.

Then, COs per million molecules ($C^M$) was calculated for each pool. The process of molecule subsampling, CO identification and $C^M$ calculation was repeated fifty times for each pool. The means ($\mu$) of all 50 $C^M$ values and their 95%

confidence intervals ($\mu \pm s$, where $s$ is $2.0096 \times \sigma/50^{0.5}$ with $\sigma$ being the standard deviation) were used to assess significant differences between samples.

**DNA methylation level estimation.** Within a recent study, we have assessed DNA methylation for *A. thaliana* Col-0[52]. Methylation level $M$ for a CO or a random interval was calculated by $M = N_{met} / N$, where $N_{met}$ is the number of reads supporting methylated cytosines at all $C$ and $G$ sites, while $N$ is the total number of reads at these sites.

**Reporting summary.** Further information on research design is available in the Nature Research Reporting Summary linked to this article.

## Data availability
Read data of all eight 10X linked-read libraries (ERS2851779-ERS2851786) and 50 whole-genome sequencing libraries (ERS2851943-ERS2851992) that support the findings of this study are available in BAM format from European Nucleotide Archive under accession number "ERP111558". All other relevant data are available upon request.

## Code availability
Custom code used for identification of recombinant molecules and frequency calculation can be found online at https://github.com/schneebergerlab/DrLink under GPL v3.0.

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

## Acknowledgements

The authors would like to thank Ian R. Henderson (Department of Plant Sciences, University of Cambridge) for providing the CO breakpoint lists, Erik Wijnker (Wageningen University) for providing seeds and Ulrike Hümann, Manish Goel, Wen-Biao Jiao, Vidya Oruganti, and Onur Dogan (Max Planck Institute for Plant Breeding Research) for their help in the greenhouse. We also would like to thank 10X Genomics for their help on setting up the *longranger* software, advice on DNA extraction, and their kind donation of library reagents to support the development of recombination identification. We thank Lutz Froenicke and the DNA Technologies Core at UC Davis for 10X library sequencing support and discussions. We also acknowledge helpful discussions with Detlef Weigel at the Max Planck Institute for Developmental Biology and Kyle Fletcher, William Palmer, and Sebastian Reyes-Chin-Wo at UC Davis. We thank Felicity Jones and Frank Chan (Friedrich Miescher Laboratory of the Max Planck Society) for inspiring the extension of this work to gametes. This work was supported by the Max Planck Society postdoctoral fellowship, and a combined grant by the Deutsche Forschungsgemeinschaft (DFG) and the Agence Nationale de la Recherche (ANR) under grant number SCHN1257/8–1 (KS), and a UC Davis Genome Center Pilot Project grant (BAR).

## Author contributions

H.S., B.A.R., and K.S. designed the project. H.S. and B.A.R. performed all analysis. B.A.R., H.S., and P.J.F. prepared the samples. B.A.R., R.B., J.F., and B.H. performed DNA extraction, quality control, library preparation, and/or sequencing. K.S., B.A.R., A.M.H., and R.M.W. supervised the project. H.S., B.A.R., and K.S. wrote the manuscript. All authors read and approved the final manuscript.
