## [Peer Review File · Nature Communications]

Reviewers' comments:

Reviewer #1 (Remarks to the Author):

In this manuscript, authors described an analytical pipeline to examine the CO landscape using the linked-read sequencing. The advantage of this methodology enables the simultaneous generation and analysis of multiple CO landscapes without generating or sequencing a single recombinant plant. Although the authors have proved the power and the benefits of this approach, I am thinking the content in this manuscript is not qualified enough to be accepted in Nature Communications. My explanations for my decision are as follows.

1. From the technical view, Nature Communications would consider any cutting-edge technology with originality. However, this work is principally based on an established method.
2. From the practical view, as described in the manuscript, the approach, the linked-read sequencing, has many inherent flaws, which may alarm the broad use in other species with the genomes not well annotated. Especially, the high degree of false positives would pose a risk on the true results. In addition, the sequencing on a pooled sample would be only achievable for male gamete, and it is unable to address the critical question about the effect of different sex on the CO distribution. Moreover, the technology of the linked-read sequencing seems unable to examine the situation about the NCO, which is also a critical question in the field of meiosis.

Reviewer #2 (Remarks to the Author):

The authors of the paper "Linked-read sequencing of gametes allows efficient genome-wide analysis of meiotic recombination" described a new method to identify meiotic crossovers (COs) using linked reads in pooled recombinants, especially in a pool of pollen DNA from a single F1 plant. The authors tested the accuracy of the method by varying the number of molecules and number of recombined genomes. The authors demonstrated that the method is highly efficient as it worked directly for F1 pollens. Overall the paper is well written and the technology is robust as it is well tested using multi-level of datasets. I have only a few minor questions.

1. It might be easier for readers to follow if the results about the association of COs with genomic features were put into an independent section. How does this uneven distribution of COs compare to the traditionally called CO hotspots?
2. In the Abstract, the authors mentioned "allows for efficient comparison of genotypic and environmental effects on recombination". it would be interesting if the authors can add some data to show that CO patterns are (or not) affected by environmental factors.

Reviewer #3 (Remarks to the Author):

The authors propose a method of enumerating meiotic crossovers (COs). They used 10X's linked-read sequencing that allows us to sequence long (typically, 10-100 kb) DNA fragments from individual homologous chromosomes in a single cell independently and to collect information on heterozygous single nucleotide variants very efficiently. 10X requires less than 1 ng of DNA, which is a major advantage over other long read sequences. With this method, they examined a large number of Arabidopsis F2 recombinants and a pool of pollen DNA from a single plant. The latter case demonstrates a merit of this approach as it outputs a landscape of COs from gametes. The approach is simple but would be useful for understanding the mechanisms for controlling recombination.

There are several major concerns on this method.

First, the technical novelty is somewhat marginal because it is a simple application of 10x's linked-read sequencing and the approach heavily relies on 10X' software programs to call SNVs and to estimate the boundaries of COs.

Second, they have confirmed several previously known properties on Arabidopsis recombination, thereby showing the accuracy of the proposal; however, they reported no serious novel findings...

Third, more than half of estimated COs are false-positive (#TP=674, #FP=1112, lines 97-98). The authors examined these FPs in details, and showed that they were likely to be random and might be ignored when we are interested in having true recombination landscapes (Fig. 3b). However, they also observed that "COs were significantly enriched in gene ends" (lines 143-4), which was found untrue and was due to many false-positive COs. It is mandatory to present how to overcome this difficulty.

Fourth, 10X linked-read sequencing has limitations in determining complete sequences of homologous chromosomes. Actually, it outputs partial information. The high rate of false-positive COs in the proposed method is likely to stem from this weakness. In the discussion, you mentioned that "long-read sequencing or single-cell DNA-seq of individual pollen would be two alternative methods, but the current costs of library preparation and sequencing are prohibitive for comparing multiple samples." (lines 253-255) Can you clarify the difference between your approach and the two alternative methods in terms of cost? Otherwise, it is hard for readers to see the tradeoff between the cost and accuracy of CO prediction.

Fifth, the method should be also accurate in removing false-negatives because understanding linkage disequilibrium is also quite essential. However, no discussions are found on the ratio of false-negatives.

Minor comments:

Line 40: "1-3" need to be a superscript.

Figures 3 and 4: Describe the meaning of purple color regions in the middle of chromosomes.

Response to Reviewer #1

General comments: In this manuscript, authors described an analytical pipeline to examine the CO landscape using the linked-read sequencing. The advantage of this methodology enables the simultaneous generation and analysis of multiple CO landscapes without generating or sequencing a single recombinant plant. Although the authors have proved the power and the benefits of this approach, I am thinking the content in this manuscript is not qualified enough to be accepted in Nature Communications. My explanations for my decision are as follows.

Comment 1. From the technical view, Nature Communications would consider any cutting-edge technology with originality. However, this work is principally based on an established method.

Response 1: We respectfully disagree with this assessment. Our approach and method allow the detection of thousands of COs in a single experiment at kb resolution without growing a single recombinant plant – this could not be and has not been done before. Our simplification in the assessment of CO landscapes enables the generation and comparison of multiple CO landscapes (e.g. from different genetic backgrounds or environments) within a single study. This was not possible before and thereby will greatly accelerate studies to understand mechanisms underlying recombination.

Comment 2. From the practical view, as described in the manuscript, the approach, the linked-read sequencing, has many inherent flaws, which may alarm the broad use in other species with the genomes not well annotated. Especially, the high degree of false positives would pose a risk on the true results. In addition, the sequencing on a pooled sample would be only achievable for male gamete, and it is unable to address the critical question about the effect of different sex on the CO distribution. Moreover, the technology of the linked-read sequencing seems unable to examine the situation about the NCO, which is also a critical question in the field of meiosis.

Response 2: We acknowledge the importance of high-quality genome assemblies for our approach; however, chromosome-level assemblies are routinely being generated for many species and soon will be available for a large diverse range of plant species. These will provide more than enough opportunities to study variation in CO landscapes across diverse taxa. Therefore, the requirement for a high-quality reference sequence is not limiting the broad applicability of our method now or in the future.

Although there were false positives, we showed that they did not obscure the true CO patterns (Main text: Fig. 3). Furthermore, we showed that the false positive rate could be reduced by loading less DNA into the Chromium controller, providing a technical solution if there are concerns about false positives interfering with the detection of true crossover patterns. In addition, the false positive issue is related to the small genome size of our study organism, Arabidopsis thaliana. Applying this method to species with larger genomes will likely result in fewer false positives. Therefore, the “cost” of false positives is relatively minor and is greatly outweighed by the benefits of having a method that can generate a kilobase resolution genome-wide recombination map based on analysis of thousands of meioses by sequencing a single sample.

We do not claim that our approach will answer every question in the meiosis field. Our approach enables a broad range of investigations on parameters influencing CO frequency during male meioses in plants. Many species of fish, insects, amphibians, reptiles, and birds can produce and lay unfertilized eggs, which can also be easily collected; this would allow comparisons between the sexes. We used male gametes because it is easy to collect them from our study organism, but the method should also work with female gametes; there have

been recent advancements in methods to collect cells from female gametophytes in plants (Luo et al, Nat Comm., 2019).

The identification of NCOs is also possible using linked-read sequencing; however, higher coverage than the current study is required for confident identification of NCOs at the single-nucleotide level. We did not assess NCOs in this study because the amount of sequencing data was limiting and we agree that this would be an interesting follow-up study.

Response to Reviewer #2

General comments: The authors of the paper "Linked-read sequencing of gametes allows efficient genome-wide analysis of meiotic recombination" described a new method to identify meiotic crossovers (COs) using linked reads in pooled recombinants, especially in a pool of pollen DNA from a single F1 plant. The authors tested the accuracy of the method by varying the number of molecules and number of recombined genomes. The authors demonstrated that the method is highly efficient as it worked directly for F1 pollens. Overall the paper is well written and the technology is robust as it is well tested using multi-level of datasets. I have only a few minor questions.

Response: We thank this reviewer for their thoughtful evaluation. We have addressed the points raised in the revised manuscript and in our responses to the comments below.

Comment 1. It might be easier for readers to follow if the results about the association of COs with genomic features were put into an independent section. How does this uneven distribution of COs compare to the traditionally called CO hotspots?

Response 1: We agree with this helpful suggestion. We have now added a new section called "Association of COs with genomic features" in main text.

In contrast to mammals, meiotic recombination in plants does not occur in well-defined hotspots, but is broadly distributed across chromosomes. Though no clear hotspots are recognizable, some broad regions do recombine more often than others. In the updated manuscript we now compare the location of 3,430 COs (identified with one of our 10X libraries) with the location of 3,320 COs aggregated from two recently published studies (Choi et al, 2016; Serra et al, 2018). For this analysis, we estimated CO frequency in 1 Mb sliding windows (with a step size of 50 Kb) along each chromosome and identified "hot" regions by merging the top 2.5% of the contiguous windows with highest CO frequencies (Fig. r1). We found 6 and 8 hot regions in our and the aggregated published datasets, respectively (Fig. r2). With the exception of the beginning of chromosome 2, these regions were found near the peri-centromeres and overlap with the two independent datasets. We now describe this analysis in the main text and Supplementary Figure 5a.

Fig. r1. Selection of CO-hot windows. CO frequency was estimated in 1 Mb sliding windows and a step size of 50 Kb along each of the chromosomes (with genome-wide normalization).

Fig. r2. CO frequencies along the chromosome. Regions with the highest CO frequencies (determined according to Fig. r1) are marked by horizontal lines in light blue (10X COs found in pooled F_2 genomes) and red (Illumina COs found in individual F_2 genomes).

Comment 2. In the Abstract, the authors mentioned "allows for efficient comparison of genotypic and environmental effects on recombination". It would be interesting if the authors can add some data to show that CO patterns are (or not) affected by environmental factors.

Response 2: *We agree that such experiments would be very interesting but they are beyond the scope of the current paper. Our goal for this paper was to develop the analytical approach and to fully evaluate its validity and accuracy. In these experiments, we tried to minimize environmental influences in order to be able to compare datasets. Several environmental factors have been reported to influence meiotic CO frequency in diverse organisms, including age, drought, nutrient stress, salinity, pathogen stress, and temperature (reviewed in Modliszewski 2017 and Fuchs 2018); however, very few have attempted to address the nature of these effects at the whole genome level and at high resolution. Our validated methodology can now be used to thoroughly and efficiently characterize previously reported and new environmental influences on CO frequency.*

Response to Reviewer #3

General comments: The authors propose a method of enumerating meiotic crossovers (COs). They used 10X's linked-read sequencing that allows us to sequence long (typically, 10-100 kb) DNA fragments from individual homologous chromosomes in a single cell independently and to collect information on heterozygous single nucleotide variants very efficiently. 10X requires less than 1 ng of DNA, which is a major advantage over other long read sequences. With this method, they examined a large number of Arabidopsis F₂ recombinants and a pool of pollen DNA from a single plant. The latter case demonstrates a merit of this approach as it outputs a landscape of COs from gametes. The approach is simple but would be useful for understanding the mechanisms for controlling recombination.

Response: We thank this reviewer for their efforts and helpful comments. We have addressed all issues as described below.

There are several major concerns on this method.

Comment 1. First, the technical novelty is somewhat marginal because it is a simple application of 10X's linked-read sequencing and the approach heavily relies on 10X' software programs to call SNVs and to estimate the boundaries of COs.

Response 1: This is not the simple application of 10X's existing pipeline. The major advance is the demonstration of the identification of thousands of COs with a single 10X library without the need for independently obtaining and analyzing sequence data for an impractical number (hundreds) of individuals. This was not possible using 10X software or its output datasets. SNP calling with the 10X software resulted in marker lists with huge levels of false positives (see response to comment 3). We developed our own novel software, workflow, and marker lists to enable the data analysis and to reduce the level of false positives to a point where they were of little to no consequence. In the revised manuscript, we now include a public link to download the software we developed (<https://github.com/schneebergerlab/DrLink>).

Comment 2. Second, they have confirmed several previously known properties on Arabidopsis recombination, thereby showing the accuracy of the proposal; however, they reported no serious novel findings...

Response 2: As described in the response to comment 2 of reviewer 2 above, the focus of this manuscript was to validate a technological advance and workflow that enables novel biological findings. This included the detailed information needed to establish the robust experimental design and the software needed to analyze the data. In response to this comment, we added a comparison of the positions of COs in pollen and F₂ genomes, which revealed 13 regions with significantly different frequencies of COs in pollen or F₂s (Fig. r3). These differences provide the starting point to investigate whether there are post-meiotic processes that influence which gametes contribute to the next generation.

Comment 3. Third, more than half of estimated COs are false-positive (#TP=674, #FP=1112, lines 97-98). They examined these FPs in details, and showed that they were likely to be random and might be ignored when we are interested in having true recombination landscapes (Fig. 3b). However, they also observed that "COs were significantly enriched in gene ends" (lines 143-4), which was found untrue and was due to many false-positive COs. It is mandatory to present how to overcome this difficulty.

Response 3: This comment was helpful because this point clearly needed better clarification. The high rate of FP referred to in this comment was based on the output of the 10X software (Longranger) and is **not** the output of our pipeline or analysis. This result was one of the

reasons why we developed our own software and method to perform the analysis (see response to comment 1).

When using our software to analyze all COs (including TP+FP) in 50-plant pools, we found that 5.16% of the COs were at gene ends, which was only marginally higher than the 4.90% of COs found at gene ends when TPs were analyzed alone. While this difference was statistically significant, it disappeared after combining all 9,695 COs (including TP+FP) in the larger F_2 pools (P200R1/2, P1250R1/2; Supplementary Figure 5c). This demonstrates that false associations of COs with genomic features due to the presence of false positive COs can be avoided as long as a sufficiently large number of COs are analyzed. It should be noted that because this false association was in part related to the small sample size, it would also likely occur in similarly small sample data sets derived from other technologies. Our paper highlights the need for large datasets, which can now be generated using our approach. We have added a discussion of these points in the revised version of the main text (“Increasing the number of genomes per library”).

Fig. r3. Regional differences (purple horizontal lines) between CO landscapes of F_2 and pollen

Comment 4. Fourth, 10X linked-read sequencing has limitations in determining complete sequences of homologous chromosomes. Actually, it outputs partial information. The high rate of false-positive COs in the proposed method is likely to stem from this weakness. In the discussion, you mentioned that “long-read sequencing or single-cell DNA-seq of individual pollen would be two alternative methods, but the current costs of library preparation and

sequencing are prohibitive for comparing multiple samples.” (lines 253-255) Can you clarify the difference between your approach and the two alternative methods in terms of cost? Otherwise, it is hard for readers to see the tradeoff between the cost and accuracy of CO prediction.

Response 4: Thank you for calling our attention to this point, as it indicated that we had failed to clarify the financial advantages using of the approach we developed. Generating pseudo-long reads with 10X libraries costs only around ~10% of the expenses of long-read sequencing technologies like PacBio and Nanopore sequencing for a similar number of molecules. We have added Supplementary Table 7, which compares estimated prices of 10X linked-read, PacBio, and Nanopore sequencing service for the same amount of data, based on the service prices at the sequencing facilities at our home institutions (Max Planck Institute for Plant Breeding Research and the University of California, Davis). The methods for single-cell DNA sequencing for the identification of COs that have been published, incur the high labor cost of sorting of individual cells or nuclei (Dreissig 2017 and Luo 2019). This has limited the throughput to about 100 cells or fewer per sample. For the same sequencing cost, the resolution of COs for single cell sequencing will be much poorer than for 10X linked read sequencing, as much higher coverage is needed to achieve kilobase-scale resolution. We have now added a more detailed discussion of the monetary and labor costs in the Discussion section.

Comment 5. Fifth, the method should be also accurate in removing false-negatives because understanding linkage disequilibrium is also quite essential. However, no discussions are found on the ratio of false-negatives.

Response 5: Based on this comment, we realized that we could have made this clearer in the manuscript and we thank the reviewer for calling it to our attention. The number of COs that can be identified within one 10X library greatly depends on the number of molecules in the library. As the probability of overlapping a CO breakpoint is the same for each molecule, the number of CO molecules depends linearly on the total number of analyzed molecules. The number of molecules is determined by the amount of DNA that is loaded in the 10X Chromium machine (and - important to note - not by the sequencing depth). As a consequence, each 10X library has a consistent number of identifiable CO molecules, based on the amount of DNA used to produce the library.

*However, the number of **distinct** CO breakpoints represented among these recombinant CO molecules depends on the number of independent CO events in the sequenced samples. If the number of CO events (i.e. a small number of plants in the pool) is much lower than the number of recombinant molecules, then most of the independent CO events will be covered by multiple CO molecules and the false negative rate will be zero. In contrast, if the number of CO events in the pool exceeds the number of CO molecules, then most of the CO molecules will be derived from different CO events, but not all of the CO events will be covered by molecules and thus some will be missed. Hence, the most important factor for the false negative rate is the number of plants and therefore the number of COs in the pool.*

We now discuss these points in the first paragraph in the section on “Increasing the number of genomes per library”.

Minor comments:

Comment 6. Line 40: “1-3” need to be a superscript.

Comment 7. Figures 3 and 4: Describe the meaning of purple color regions in the middle of chromosomes.

Response 6&7: Thank you for the comments. We have revised the text accordingly.

REVIEWERS' COMMENTS:

Reviewer #1 (Remarks to the Author):

Thanks the authors make efforts to improve the manuscript. I agree that by performing the methods as described in the manuscript would be beneficial to examine multiple CO landscapes within a single study. However, it would be definitely required to test this conclusion with experiments, such as e.g. from different genetic backgrounds or environments. Otherwise, the novelty or merit of this manuscript is limited.

Reviewer #2 (Remarks to the Author):

The authors have addressed my previous concerns in their revision. I have a minor suggestion that a citation regarding the environmental factors influencing meiotic CO frequency should be added since the authors mentioned the environmental effects in the Abstract.

Reviewer #3 (Remarks to the Author):

The authors answered all the questions and comments clearly. I appreciate their efforts on maximizing the capability of 10X by reducing numerous false-positive calls...

In the abstract, to attract a wider readership, you can mention the 13 regions with significantly different frequencies of COs.

In the newly added section entitled "Increasing the number of genomes per library," they clarified how to overcome the limitations of the proposed method and to accelerate its performance. In the response to my comments, they mentioned the difficulty in defining "false-negatives," which is informative and should be also included into the new section for better understanding of readers.

Response to Reviewer #1

General comments: Thanks the authors make efforts to improve the manuscript. I agree that by performing the methods as described in the manuscript would be beneficial to examine multiple CO landscapes within a single study. However, it would be definitely required to test this conclusion with experiments, such as e.g. from different genetic backgrounds or environments. Otherwise, the novelty or merit of this manuscript is limited.

We viewed the main scope of this study to be the development and evaluation of the method. In particular we have described our extensive tests and validations of the approach including sequencing of pools of genomes, which we also sequenced individually. This allowed us to assess the performance of the method at a level which is usually not possible. We then (for the first time) applied the method to a pool of pollen DNA and analyzed the recombination landscape in the pollen in contrast to the recombination landscapes that we assessed in the offspring genomes. After all of this, we now can safely estimate that it is feasible to compare multiple genetic backgrounds or environments in a single study.

Response to Reviewer #2

General comments: The authors have addressed my previous concerns in their revision. I have a minor suggestion that a citation regarding the environmental factors influencing meiotic CO frequency should be added since the authors mentioned the environmental effects in the Abstract.

We had included several citations documenting the influence of environmental stress on meiotic recombination in the Introduction (see paragraph 1, page 3), but we realized based on this comment that we should have included more about this subject. We have now added this in the Discussion (see paragraph 1, page 13).

Response to Reviewer #3

General comments: The authors answered all the questions and comments clearly.

I appreciate their efforts on maximizing the capability of 10X by reducing numerous false-positive calls...

Thank you.

In the abstract, to attract a wider readership, you can mention the 13 regions with significantly different frequencies of COs.

We agree that this is an important result that we could achieve with our new method. However, we would prefer to focus the abstract on the actual research goal, which is the development and evaluation of a new method in order to emphasize the broad application possibilities of this method.

In the newly added section entitled "Increasing the number of genomes per library," they clarified how to overcome the limitations of the proposed method and to accelerate its performance. In the response to my comments, they mentioned the difficulty in defining "false-negatives," which is informative and should be also included into the new section for better understanding of readers.

We have amended this section to include the requested information.